# Early Cretaceous *Keteleerioxylon* Wood in the Songliao Basin, Northeast China, and Its Geographic and Environmental Implications

**DOI:** 10.3390/biology11111624

**Published:** 2022-11-07

**Authors:** Xiao Shi, Yuewu Sun, Fanli Meng, Jianxin Yu, Zilie Lan

**Affiliations:** 1Department of Geology, College of Earth Sciences, Jilin University, Changchun 130061, China; 2International Center of Future Science, Dinosaur Evolution Research Center, Jilin University, Changchun 130012, China; 3Research Center of Paleontology and Stratigraphy, Jilin University, Changchun 130026, China; 4Bioarchaeology Laboratory, Jilin University, Changchun 130021, China; 5School of Earth Sciences, State Key Laboratory of Biogeology and Environmental Geology, China University of Geosciences, Wuhan 430074, China

**Keywords:** *Keteleerioxylon changchunense* sp. nov., *Keteleeria*, geography, environment, Cretaceous

## Abstract

**Simple Summary:**

*Keteleeria* is a small group of Pinaceae, now only distributed in east and southeast Asia, but which was widely distributed in Asia, Europe, and North America in the northern hemisphere during the Late Mesozoic and Cenozoic periods. A new *Keteleeria*-like wood fossil, *Keteleerioxylon changchunense* Shi, Sun, Meng et Yu sp. nov., was described in the Early Cretaceous strata about 110 million years ago in Changchun, Jilin Province, Northeast China. The growth rings of wood contain rich palaeoecological and paleoclimatic information. Quantitative analysis of growth rings revealed that the new species is evergreen with leaf longevity of 1–3 years. The growth rings of *Keteleerioxylon changchunense* indicate that the climate seasonality was pronounced in the Songliao Basin during the Early Cretaceous period. By reviewing *Keteleeria* and closely related fossil taxa, we depict the probable migration route of *Keteleeria*. The oldest *Keteleeria*-like wood was found in the middle Jurassic period in Svalbard, Norway. They were distributed in both the middle and the high latitudes during the Late Jurassic–Cretaceous global warming time, while during the Paleogene and Neogene cooling times, the *Keteleeria*-like wood became scarce, especially in the Quaternary Glaciation, and until now, they were strictly restricted to the subtropical and tropical regions of east and southeast Asia.

**Abstract:**

The extant *Keteleeria* is endemic to east and southeast Asia, while *Keteleeria*-like trees were widely distributed in the northern hemisphere in Earth’s history. In this paper, we reported a novel wood fossil of *Keteleerioxylon changchunense* Shi, Sun, Meng et Yu sp. nov., collected from the middle part of the Yingcheng Formation, Yingcheng Coal Mine, Changchun City, Jilin Province, northeast China. The quantitative growth-ring analyses of *K. changchunense* indicate that it was evergreen with a leaf longevity of 1–3 years, which is consistent with the foliar retention of extant *Keteleeria*. Its high ring-markedness index (RMI) indicates that the climate seasonality was pronounced during the Early Albian period in the Songliao Basin, northeast China. The fossil records of *Keteleeria* and closely related taxa indicate that this group might have originated from high latitudes in the northern hemisphere, then spread and migrated southward during the Late Jurassic and Cretaceous periods, gradually decreased in the Cenozoic period, and so far only survives in east and southeast Asia.

## 1. Introduction

*Keteleeria* was firstly described as a genus in 1866, named by Carrière [1]. The genus *Keteleeria* Carrière includes ten extant species and two variants, which are endemic to east and southeast Asia and only found in southern China (from Qinling Mountain to Hainan Island), northern Laos, and southern Vietnam [2] (Figure 1). *Keteleeria* is confined to humid, moderately warm (subtropical) areas with relatively low mountains (200–3000 m above sea level) [3,4,5,6].

Fossil woods with similar anatomy to extant *Keteleeria* were first described in the Lower Cretaceous of Franz Josef Land with the name *Keteleerioxylon* [7]. Although *Keteleeria* and *Abies* separated in the Paleocene based on molecular clock estimation [8], the oldest representatives of *Keteleeria*-like (the fossil woods, leaves, and reproductive organs showing similarities to the genus *Keteleeria*, including *Keteleerioxylon*, *Keteleeria*, and *Protopiceoxylon* or *Pinoxylon*) wood might have been in the Middle Jurassic [9]; most fossils reLated to the *Keteleeria* were discovered from the Late Jurassic to Cenozoic strata of Asia, Europe, and North America in the northern hemisphere (e.g., [10,11,12]).

In this study, we describe two new silicified wood specimens belonging to the genus *Keteleerioxylon* from the early Cretaceous sediments in the Songliao Basin, northeast China, and named *Keteleerioxylon changchunense* sp. nov. Quantitative growth-ring analyses were made to understand ecological and climatic indications. Meanwhile, the geological and geographical distributions of *Keteleeria*-like fossil woods are summarized in order to depict the relationship between its migration and paleoclimate changes during the Cretaceous and Cenozoic times.

## 2. Materials and Methods

### 2.1. Materials

Two silicified woods (registered nos. JLJY-01 and JLJY-02) were collected from the middle section of the Yingcheng Formation, Yingcheng Coal Mine (44°9′47.53″ N, 125°54′53.92″ E), Changchun City, Jilin Province, northeast China (Figure 2). The location is situated at the southeast margin of the Songliao Basin, the largest Mesozoic–Cenozoic terrestrial oil-bearing basin in northeast Asia. The Lower Cretaceous succession in the Songliao Basin is composed of the Huoshiling, Shahezi, Yingcheng, and Denglouku formations in ascending order [13]. The Yingcheng Formation, dominated by medium-acid volcanic, volcaniclastic rocks, and coal-bearing deposits, can be divided into three parts. The middle section is a set of sedimentary rocks with coal layers yielding spores and pollen, leaves, woods, and insects.

Detailed isotopic dating of volcanic rocks indicates that the age of the middle section is Late Aptian–early Albian, between 115.2 ± 0.4 Ma and 110.0 ± 2.0 Ma [14,15]. Based on palynological data, due to the presence of *Tricolpites* sp. in the middle part of the Yingcheng Formation where the present silicified wood specimens were collected, the member is considered earliest Albian in age [16,17,18,19].

The specimen JLJY-01 is 45.5 cm long with a maximum diameter of 13.5 cm. The specimen JLJY-02 is 29.4 cm long and 11.0–21.6 cm in diameter. They are gray to black in color with well-preserved secondary xylem. However, no pith or primary xylem is preserved.

All the specimens and slides are housed in the Research Center of Paleontology and Stratigraphy, Jilin University, Changchun, China.

### 2.2. Methods

To investigate the anatomical characteristics of those silicified woods, microscopic slides of the transverse, radial, and tangential sections were made in the State Key Laboratory of Biogeology and Environmental Geology, China University of Geosciences (Wuhan). The slides were observed using a microscope (Nikon AZ100M) with a digital camera (Nikon DS-Ri2) in the International Center of Future Science, Dinosaur Evolution Research Center, Jilin University. Images were processed by the software Adobe Photoshop CS6 (Version 13. San Jose, CA, USA: Adobe Inc., licensed to Jilin University). 

For the quantitative growth ring analysis, the radial diameters of tracheids were measured, and the percentage of diminution, percentage of latewood, ring markedness index (RMI), and percentage of skews of CSDM (cumulative sum of the deviation from mean diameter) curves were calculated following the method of Falcon-Lang [20].

## 3. Results

ORDER Coniferales.

GENUS *Keteleerioxylon* I.A. Shilkina, 1960.

Type species: *Keteleerioxylon arcticum* I.A. Shilkina, 1960.

*Keteleerioxylon changchunense* Shi, Sun, Meng et Yu sp. nov.

Holotype: Specimen JLJY-02 (Figure 3A); Slides JLJY-02 a1, JLJY-02 a2, JLJY-02 b1, JLJY-02 b2, JLJY-02 c1, JLJY-02 c2.

Repository: Research Center of Paleontology and Stratigraphy, Jilin University, Changchun, China.

Type locality: Yingcheng Coal Mine, Changchun City, Jilin Province, China.

Stratigraphic horizon and age: The middle section of the Yingcheng Formation, earliest Albian (Early Cretaceous).

Etymology: The specific name is derived from Changchun where the specimens were collected.

Diagnosis: Growth rings distinct. Pits on radial walls of tracheids uniseriate to triseriate. Pits circular; uniseriate pits scattered, rarely contiguous slightly; biseriate pits mostly opposite, occasionally alternate; triseriate in a row. Crassulae present. Pits on tangential walls of tracheids absent. Transverse walls of axial parenchyma cells smooth. Rays, (1)6–11(37) cells high, uniseriate, sometimes with one to eight layers of biseriate cells. Horizontal and end walls of ray cells distinctly pitted (Abientineentüpfelung). Ray marginal cells, resembling ray tracheids, with scattered uniseriate pits. Pits, of taxodioid-cupressoid type, 1–3(6) per cross-field, 5–10 μm in diameter. Resin canals, vertical, normal, surrounded by six to eleven thick-walled epithelial cells in a ring. Horizontal resin canals absent.

Description: The two specimens show similar characteristics, but the features of the transverse section are clearer in specimen JLJY-02. Thus, the description here is mainly based on the specimen JLJY-02. Only the secondary xylem was preserved in the two specimens.

The homoxulous pycnoxylic wood consists of tracheids, rays, axial parenchyma cells, and epithelial cells of resin canals. The growth rings are distinct, 0.79–2.85 mm wide, with distinct and relatively straight boundaries (Figure 3B,D). The early-late wood transition is abrupt. In the transverse section, the early wood tracheids are large, thin-walled, and mainly rectangular (sometimes polygonal or circular). The latewood tracheids are thick-walled, and radially compressed, and lumens almost disappeared near the growth-ring margin. The radial tracheid lumens are 0–89 μm in diameter. Intercellular space is absent. Xylem rays mainly consist of uniseriate cells, and 1–10 seriates of tracheids in between.

Pits on the radial walls are borded, circular in shape, (11)12–17(20) μm in diameter. In the late woods, they are uniseriate, partly biseriate, and scattered, rarely slightly contiguous, with an included aperture (circular or elliptical). In the early woods, they are mostly biseriate or triseriate, occasionally uniseriate (Figure 3F,G,H and Figure 4A). When biseriate, the pits are mainly opposite, occasionally they show a tendency toward alternate arrangement (Figure 3G,H); when triseriate, the pits arrange in a row (Figure 4A). The pits are (16)18–22(25) μm in diameter, with circular or elliptical apertures. Crassulae are present (Figure 3F,H and Figure 4A). Axial parenchyma cells are rare, and `their transverse walls are smooth (Figure 4C).

Rays are (1)6–11(37) cells in height, uniseriate, occasionally with one to eight layers of biseriate cells (Figure 4G,H). In the radial section, ray cells are brick-shaped, 19–25 μm in height, and 50–129 μm in length. The horizontal and end walls of the ray parenchyma cells are pitted (Figure 4B,I). In the tangential section, ray cells are 10–28 μm in width, elliptical, circular, or rectangular, but rounded-triangular in marginal ray cells. Marginal ray cells resemble those ray tracheids with scattered uniseriate pits (Figure 4F).

There are 1–3(6) pits per cross-field, of taxodioid-cupressoid type, 5–10 μm in diameter. (Figure 4D,E). Two or three pits arrange in a row, while four to six pits are in a diffuse arrangement.

Vertical resin canals are present only (Figure 3E). Resin canals are circular or oval, radially elongated, 47–206 μm in diameter, and single or in pairs (Figure 3B,C). They are lined with six to eleven thick-walled epithelial cells in a ring and situated in both early and latewood. Horizontal resin canals are absent.

Comparison: The fossil woods described here are characterized by the presence of (only) vertical normal resin canals with thick-walled epithelial cells, axial parenchyma, and the marginal cells of rays resembling ray tracheids. They closely resemble the extant genus *Keteleeria* [11].

The presence of (only) vertical normal resin canals with thick-walled epithelial cells and the absence of horizontal resin canals are unique characteristics of the extant *Keteleeria* [7,11,21,22,23,24,25]. Sole vertical resin canals may also occur in the wood of some taxa of the Taxodiaceae and Cupressaceae. However, these canals are traumatic, relatively large, with irregular outlines, and without typical epithelial cells that form a continuous lining around a resin canal [11].

The absence of true ray tracheids is also an important characteristic of *Keteleeria* [22]. The marginal ray cells of *Keteleeria* have a slightly greater number of pits on the radial walls and undulate slightly convex external walls. The phenomenon can be also observed in the wood of *Abies* and *Pseudolarix* [22]. However, the two taxa do not possess normal resin canals.

The presence of axial parenchyma is also the characteristic of *Keteleeria*, although it is scarce [21,22,23].

Shilkina proposed the generic name *Keteleerioxylon* for the fossil woods with the anatomical characteristics of the extant *Keteleeria* [7]. *Pinoxylon* (Knowlton) Read or *Protopiceoxylon* Gothan (the generic name is controversial) shows similar characteristics with *Keteleerioxylon* in possessing sole vertical resin canals [25,26,27,28,29]. As clearly explained by Philippe and Bamford, *Pinoxylon* or *Protopiceoxylon* shows araucarian, or araucarian and abietinean pitting on the radial walls of tracheids, while the pits on the tracheid radial walls of *Keteleerioxylon* are abietinean, or slightly alternate when biseriate [29]. In the wood of *Keteleeria*, the tracheid pitting in radial walls is abietinean, but occasionally shows a tendency toward “alternate” [30]. The radial pitting of the studied fossil wood is quite like extant *Keteleeria*. Thus, the present specimen should be assigned to the genus *Keteleerioxylon*.

The present fossil woods differ from the extant species of *Keteleeria* in having uniseriate to triseriate pits on the radial walls of tracheids, in lacking pits on tangential walls of tracheids, and in possessing taxodioid and cupressoid cross-field pits (Table 1).

The fossil wood genus *Keteleerioxylon* was established from the materials collected from the Early Cretaceous Franz Josef Land, Russia by Shilkina [7]. Until now, four species of *Keteleerioxylon* have been described. *K. arcticum* Shikkina, 1960 differs from *K. changchunense* in having uniseriate to biseriate pits on the radial walls of tracheids, in possessing pits on tangential walls of tracheids, in lacking marginal ray cells of the ray tracheid type, and in having fewer pits per cross-field [7]. *K. fokinii* Shilkina, 1986 was found in the Early Cretaceous Kirovsk Region of Russia [34]. *K. fokinii* differs from the present woods in having uniseriate to biseriate pits on the radial walls of tracheids, in the absence of crassula, and in the taxodioid cross-field pits. *K. primoryense* Blokhina, 2000 from the Oligocene–Miocene of Primorye differs from *K. changchunense* in having uniseriate to biseriate pits on the radial walls of tracheids, in possessing pits on tangential walls of tracheids, and in the cupressoid cross-field pits [35]. *K. kamtshatkiense* Blokh. et Afonin, 2006, from the Lower Cretaceous of the Kamchatka Peninsula, Russia, is different from the woods under the description, in having uniseriate to biseriate pits on the radial walls of tracheids, in possessing pits on tangential walls of tracheids, and in the absence of crassula [11]. Jiang et al. reported that the *Keteleeria liaoxiense* and *Keteleeria* sp. were discovered from the Middle Jurassic Liaoning Province, but they did not give a description [36].

There are many woods that were identified as *Keteleeria*. *Keteleeria mabetiensis* Watari, 1941 was collected from the Lower Miocene of Japan [37,38]. *Keteleeria mabetiensis* differs from *Keteleerioxylon changchunense* in possessing pits on tangential walls of tracheids, in knotty transverse walls of axial parenchyma, and taxodioid and piceooid cross-field pits. *Keteleeria zhilinii* Blokhina et Bondarenko, 2005 was first discovered in the Pliocene of Primorye, Russia [10]. *K. zhilinii* differs from the woods under study in having uniseriate to biseriate pits on the radial walls of tracheids, in having pits on tangential walls of tracheids, and in the taxodioid cross-field pits. Yang et al. discovered a silicified wood belonging to the genus *Keteleeria* in the Late Cretaceous of Henan, China [39]. They named these woods after the extant species “*Keteleeria fortunei*”. “*Keteleeria fortunei*” *i*s different from *Keteleerioxylon changchunense* in having uniseriate to biseriate pits on the radial walls of tracheids and in the absence of axial parenchyma. *Keteleeria* sp. from the early Holocene of Hubei, China, differs from *Keteleerioxylon changchunense* in having uniseriate to biseriate pits on the radial walls of tracheids and in the taxodioid cross-field pits [40]. *Keteleeria* sp. from Guangdong, China differs from the woods under study in having uniseriate to biseriate pits on the radial walls of tracheids, in lower rays, and in having fewer pits per cross-field [12].

For the details of each fossil record, please see Appendix A.

## 4. Discussion

### 4.1. Paleoecological and Paleoclimatic Implications

Quantitative analysis of growth rings can be used to detect whether the conifer species is evergreen or deciduous, and how long the leaf longevity is [20,41]. Through the measuring of radial diameters of five adjacent files of tracheid cells, four parameters were calculated, including (1) percentage of latewood, (2) percentage of cell diminution in a ring increment, (3) RMI, and (4) percentage of skews of CSDM curves (Table 2, Appendix A).

Deciduous conifers have dominantly left-skewed CSDM or symmetrical curves, whereas evergreen conifers mainly have right-skewed CSDM curves. The CSDM curves of *Keteleerioxylon changchunense* are from 0 to +20.00% (mean percentage of skew +8.77%), right-skewed, suggesting that this species was evergreen (Figure 5, Table 3).

The percentage of latewood in *Keteleerioxylon changchunense* is 40.00–50.00%, with a mean of 45.62%; and the percentages of the CSDM curves skews range from 0–20.00% (mean 8.77%). These two parameters are quite close to those of *Pinus sylvestris* and *Picea abies*, while the percentage of cell diminution (89.39–97.50%, mean 95.45%) and the RMI (38.92–48.75%, mean 43.51%) are higher than those of *Larix decidua* (Table 2). Therefore, the leaf retention time of *Keteleerioxylon changchunense* is considered to be 1–3 years, the most likely approach to the extent of *Keteleeria* [42].

The CSDM curves are widely used for identifying evergreen or deciduous habits. However, an exception comes from the Late Pleistocene *Keteleeria* sp. of Guangdong, China, as its CSDM curves showed both left and right skews [12]. However, the radial cells in three growth rings of *Keteleeria* sp. are less than 15; this might result from false rings triggered by the East Asian monsoon. Therefore, we still think the right skew of CSDM curves is reliable in recognizing evergreen trees like *K. changchunense*.

The earliest Albian climate is regarded as a greenhouse with high atmospheric carbon dioxide concentration [19]. The global temperature in the Aptian and Albian periods would have been much higher than today [43], and the Arctic area was covered by evergreen broadleaf mixed with deciduous conifer forests in the Late Cretaceous period [44]. The growth ring RMI is considered as an indicator of the climatic seasonality intensity [20]. *Keteleerioxylon changchunense* shows very marked growth rings (high RMI) (Table 3). This might indicate that the climate seasonality was likely to be strong in the earliest Albian period in the Songliao Basin, Northeast China.

### 4.2. Phytogeographical Distribution

As mentioned above, *Keteleerioxylon* and *Protopiceoxylon* or *Pinoxylon* show great similarity to the extant *Keteleeria*. According to Blokhina et al., there is a distinct possibility that they are the remote ancestors of *Keteleeria*. Although extant *Keteleeria* is endemic to East and Southeast Asia, the *Keteleeria*-like fossils (including *Keteleeria*, *Keteleerioxylon*, and *Protopiceoxylon* or *Pinoxylon*) had been widely distributed in Europe, North America, and Asia in the northern hemisphere since the Middle Jurassic period (Table 4).

The earliest record of a *Keteleeria*-like fossil was discovered in Middle Jurassic Svalbard, Norway. The Late Jurassic-Early Cretaceous *Keteleeria*-like trees are widely distributed from western Liaoning, China to the Arctic areas (Figure 6). Their paleolatitudes range from 36° N to 85° N (Figure 6). This indicates that the climate might be warm and suitable for these subtropical thermophilic *Keteleeria*-like trees in the Late Jurassic-Early Cretaceous. In the Late Cretaceous, *Keteleeria*-like trees were still persistently distributed in the middle and high latitudes of the Russian far east and northeast China. After the K-Pg mass extinction, records of *Keteleeria* fossils were rare in the Paleocene [56,63]. In the Eocene, *Keteleeria* migrated to middle latitudes [6,64]. This migration was contemporary with the Paleocene–Eocene thermal maximum [88,89]. During the Oligocene and Miocene, *Keteleeria* was widely distributed in the mid-latitudes of North America, Europe, and east Asia in the northern hemisphere. From the Pliocene on, *Keteleeria* started to retreat southward from Primorye, Russia to the subtropical and tropical monsoon climate regions of South China and Southeast Asia in the middle and low latitudes (~13.8° N–35.5° N) in the Quaternary. Therefore, the palaeogeographical distribution of *Keteleeria*-like was closely related to the paleoclimate changes (Figure 6).

## 5. Conclusions

In this study, the species *Keteleerioxylon changchunense* Shi, Sun, Meng et Yu sp. nov. is described for the first time in the Late Early Cretaceous deposits of Changchun, Northeast China. The quantitative growth ring analysis of the two specimens shows that the species is evergreen, with the foliar long around 1–3 years. This is quite consistent with the habits of extant *Keteleeria*. The presence of very marked growth rings indicates that it lived in a seasonal climate during the earliest Albian period in the Songliao Basin, northeast China. The palaeogeographical distribution of the *Keteleeria*-like fossils indicates that they were distributed in both the middle and the high latitudes during the Late Jurassic–Cretaceous global warming time, while they retreated to the middle latitude in the Cenozoic due to global cooling, and they have only survived in east and southeast Asia after the Quaternary glaciations until now.

## Figures and Tables

**Figure 1 biology-11-01624-f001:**
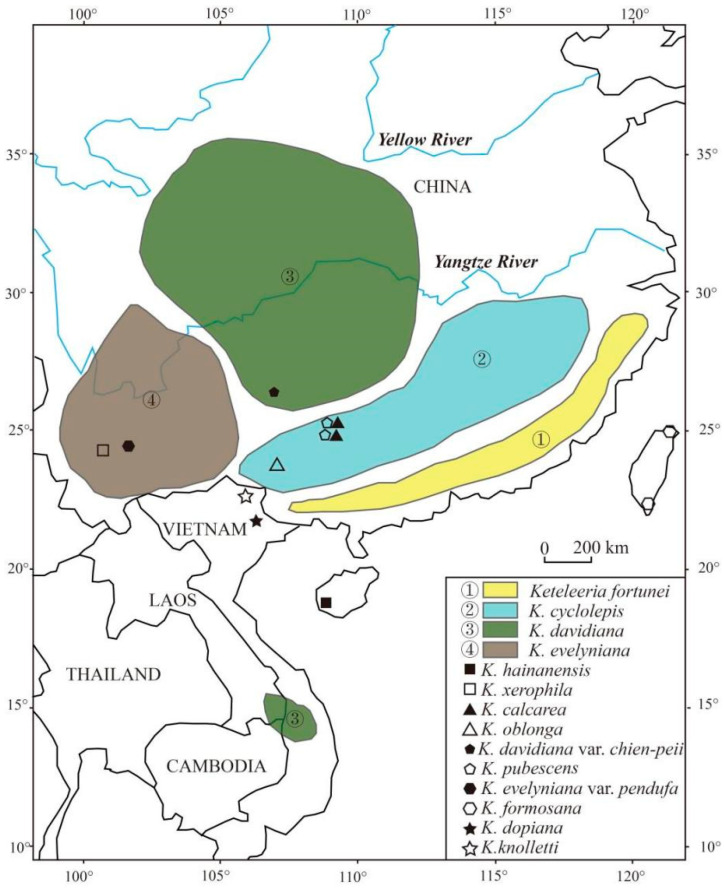
Distribution of the extant *Keteleeria* species and variants in the world (Adapted with permission from Ref. [2]).

**Figure 2 biology-11-01624-f002:**
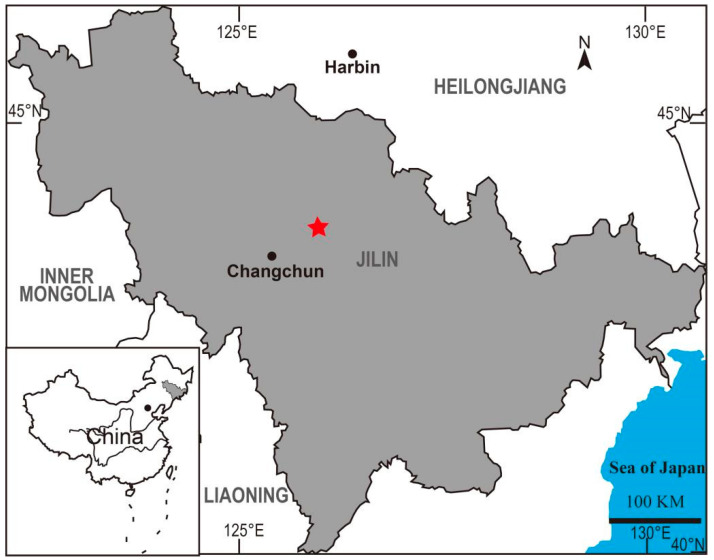
Location of *Keteleerioxylon changchunense* Shi, Sun, Meng et Yu sp. nov. in the Yingcheng Coal Mine (red star), Changchun City, Jilin Province, Northeast China.

**Figure 3 biology-11-01624-f003:**
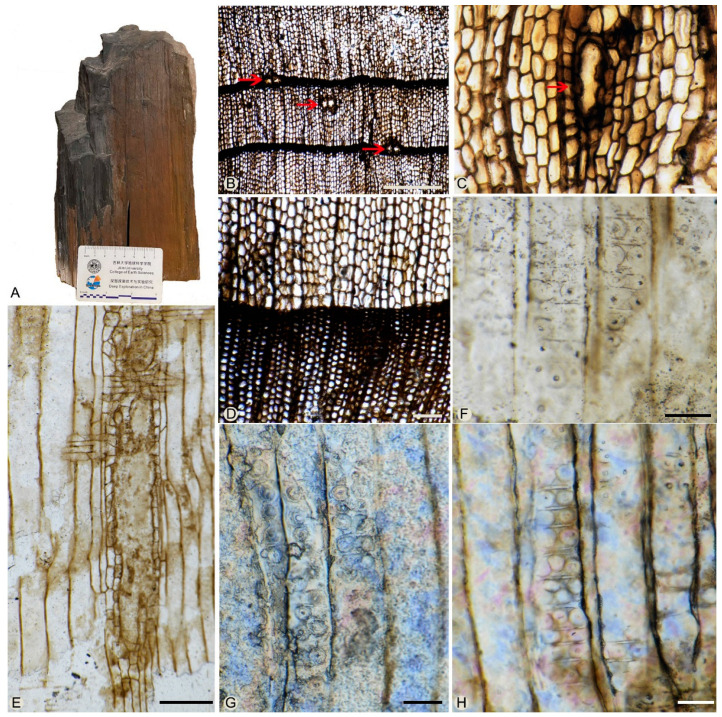
*Keteleerioxylon changchunense* Shi, Sun, Meng et Yu sp. nov. (**A**). General view of the specimen JLJY-02. (**B**). Transverse section showing the resin canals (red arrows) and growth ring boundaries. (**C**). Close-up of the resin canals (red arrow). (**D**). Close-up of the growth ring boundary. (**E**). Radial section showing the vertical resin canal. (**F**). Radial section showing the biseriate pits on the radial walls of tracheids and the crassulae. (**G**,**H**). Radial section showing the opposite and alternate biseriate pits and triseriate pits and the crasulae on the radial walls of tracheids. Scale bar: A: the longer scale mark in the card is 1 cm; B = 1 mm; C, D, E = 200 μm; F, G, H = 50 μm.

**Figure 4 biology-11-01624-f004:**
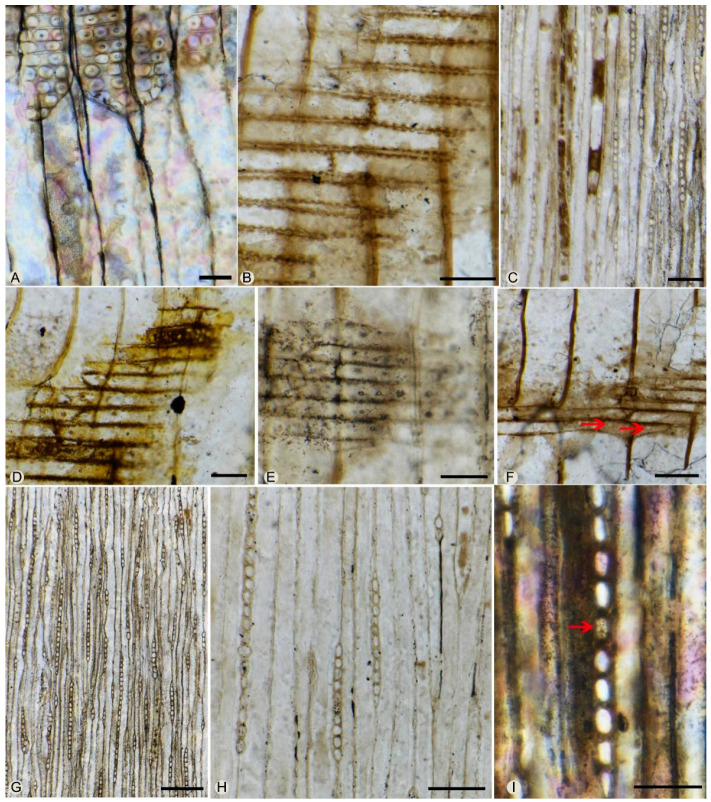
*Keteleerioxylon changchunense* Shi, Sun, Meng et Yu sp. nov. (**A**). Radial section showing the opposite and alternate biseriate pits and triseriate pits and the crasulae on the radial walls of tracheids. (**B**). Radial section showing pitted horizonal and end walls of the ray parenchyma cells. (**C**). Tangential section showing the axial parenchyma cell. (**D**,**E**). Radial section showing taxodioid-cupressoid type cross-field pits. (**F**). Radial section showing the marginal ray cells resembling ray tracheids with scattered uniseriate pits (red arrows). (**G**). Tangential section showing uniseriate, partly biseriate rays. (**H**). Close-up of the rays showing uniseriate rays. (**I**). Tangential section showing the pitted end wall of ray parenchyma cell (red arrow). Scale bar: A, B, D, E, F, I = 50 μm; C, H = 100 μm; G = 200 μm.

**Figure 5 biology-11-01624-f005:**
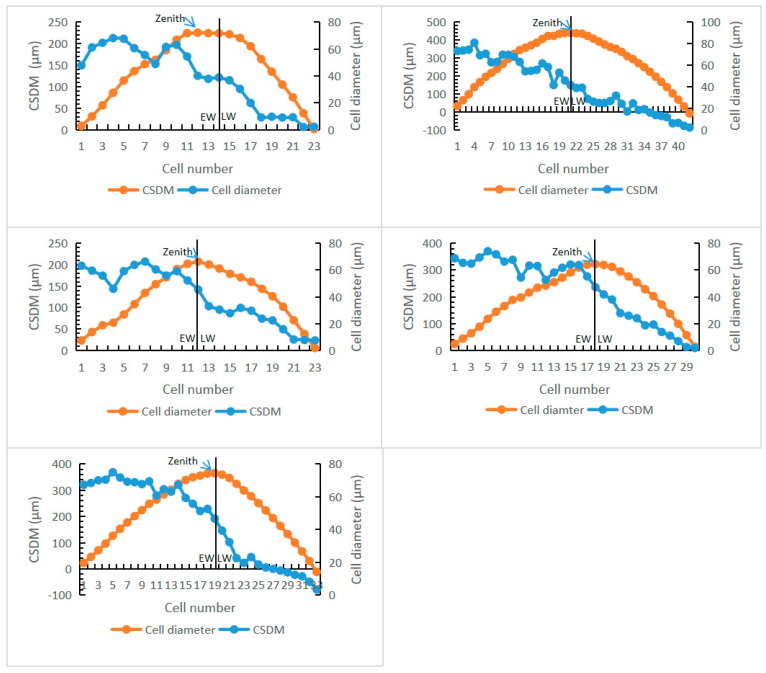
The CSDM curves of five growth rings and cell diameters of growth ring increment. EW: Earlywood, LW: latewood. The arrows indicate the zeniths of CSDM curves. When the abscissa value of the zenith is greater than half of the cell number, the curve is right-skewed; on the contrary, it is left-skewed.

**Figure 6 biology-11-01624-f006:**
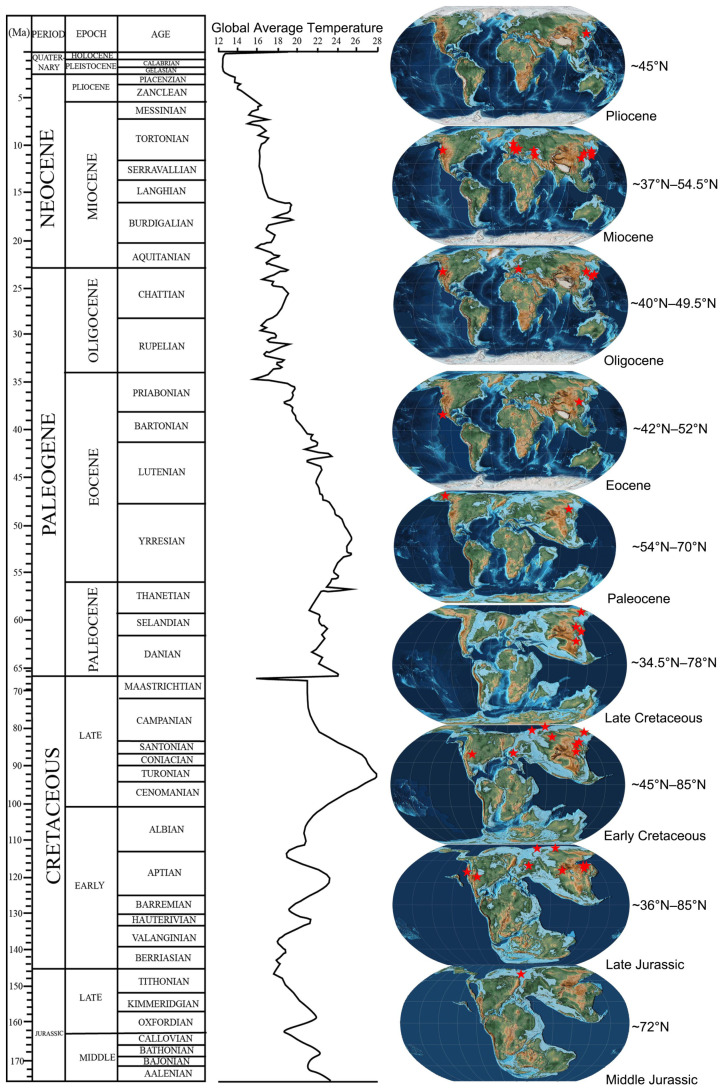
Geographical and geological distribution of *Keteleeria*-like fossils (red stars) in the world (global average temperature data from [90]).

**Table 1 biology-11-01624-t001:** Comparison of wood anatomical characteristics of *Keteleerioxylon changchunense* sp. nov. and woods of extant *Keteleeria*.

Anatomical Characters	*Keteleeria davidiana* (Bertr.) Beissner [11,22,31,32,33]	*Keteleeria evelyniana* Masters [11,22]	*Keteleeria fortunei* (Andr. Murray) Carr. [11,21,22,32]	*Keteleerioxylon changchunense* sp. nov.
Pits on radial walls of tracheids	Uniseriate to tetraseriate	Uniseriate to biseriate	Uniseriate to biseriate	Uniseriate to triseriate
diameter of pits, μm	(10)12–15(16)	15–21	12–15(16)	(11)12–17(20)
Pits on tangential walls of tracheids	Present	Present	Present	Absent
Crassulae	Present	Present	No data	Present
Uniseriate rays: height (in cells)	1–30(34)	1–30	1–40	(1)6–11(37)
number of biseriate layers	1–8	Uncommon	Present	1–8
marginal ray cells of the ray tracheid type	Present	Present	Present	Present
Transverse walls of axial parenchyma	Knotty	?	Knotty	Smooth
Number of epithelial cells in vertical resin canals	6–7(8–11)	6	6–8	6–11
Pitting on cross-fields: number of pits	1–3(6)	1–4	1–4(6)	1–3(6)
diameter of pits, μm	(4)5–8(12)	No data	4–6	5–10
type of pitting	Taxodioid, piceoid, and pinoid	taxodioid	taxodioid	Taxodioid and cupressoid

**Table 2 biology-11-01624-t002:** Results of the quantitative growth ring analysis of *Keteleerioxylon changchunense* sp. nov.

Ring Number	Percentage Latewood	Percentage Diminution	Ring Markedness Index (RMI)	Percentage Skews
Ring A	48.00%	97.06%	46.42%	+4.35%
Ring B	50.00%	97.50%	48.75%	0%
Ring C	48.00%	89.39%	42.76%	+4.35%
Ring D	40.00%	97.30%	38.92%	+20.00%
Ring E	42.00%	96.00%	40.72%	+15.15%
Averages	45.60%	95.45%	43.51%	+8.77%

**Table 3 biology-11-01624-t003:** Comparison of the quantification of ring markedness parameters for *Keteleerioxylon changchunense* sp. nov. with the five extant taxa (data from [20]).

Tree Habits	Species	Leaf Retention Time (in years)	Percentage of Latewood (%)	Percentage Diminution (%)	Ring Markedness Index (RMI) (%)	Range of Percentage Skews (Mean Value) (%)
Deciduous	*Larix decidua*	<1 year	50.00–54.83	71.55–85.91	35.77–44.36	−40.0 to +7.7 (−6.8)
Evergreen	*K. changchunense*	1–3 years	40.00–50.00	89.39–97.50	38.92–48.75	0.0 to +20.0 (+8.8)
	*Pinus sylvestris*	1–3 years	41.03–50.00	70.53–77.28	31.56–35.26	−9.1 to +17.9 (+5.2)
	*Picea abies*	3–5 years	25.93–44.19	74.02–84.03	19.90–35.42	0.0 to +38.2 (+12.0)
	*Cedrus libani*	3–6 years	30.77–39.58	62.33–72.06	20.22–24.68	+35.7 to +42.9 (+39.0)
	*Araucaria araucana*	3–15 years	10.00–22.50	28.67–51.79	3.17–10.35	+55.0 to +80.0 (66.7)

**Table 4 biology-11-01624-t004:** Geological distribution of *Keteleeria*-like fossils (excluding the species without detailed description).

Species	Location	Age	Type of Fossil	References
*Protopiceoxylon wordii* Walton	Svalbard, Norway	Middle Jurassic	Wood	Walton, 1927 [9]
*Protopiceoxylon articum* Seward	Franz Josef Land, Russia	Late Jurassic	Wood	Seward, 1904 [45]
*Protopiceoxylon resiniferous* Medlyn et Tidwell	Utah, USA	Late Jurassic	Wood	Medlyn and Tidwell, 1979 [46]
*Protopiceoxylon canadense* Medlyn et Tidwell	British Columbia	Late Jurassic	Wood	Medlyn and Tidwell, 1979 [46]
*Pinoxylon* (*Protopiceoxylon*) *dacotense* knowlton	Liaoning Province, China; South Dakota, USA; Sakhalin, Russia	Late Jurassic; Early Cretaceous (?); Late Cretaceous	Wood	Jiang et al., 2008 [47]; Knowlton, 1900 [25]; Nishida and Nishida, 1995 [48]
*Protopiceoxylon extinctum* Gothan	Hebei Province, China; Svalbard, Norway	Late Jurassic; Early Cretaceous (?)	Wood	Gothan, 1907 [27]; Mathew and Ho, 1945 [49]; Sze, 1963 [50]
*Protopiceoxylon xinjiangense* Wang, Zhang et Saiki	Xinjiang, China	Late Jurassic	Wood	Wang et al., 2000 [51]
*Protopiceoxylon* (*Pinoxylon*) *yabei* (Shimakura) Sze	Jilin, China	Late Jurassic	Wood	Shimakura, 1936 [52]; Mathews and Ho, 1945 [49]; Sze, 1963 [50]
*Keteleerioxylon fokinii* Shilkina	Kirovsk Region, Russia	Valanginian (Early Cretaceous)	Wood	Shilkina, 1986 [34]
*Protopiceoxylon edwardsi* Stopes	Greensands, England	Aptian (Early Cretaceous)	Wood	Stopes, 1915 [53]
*Protopiceoxylon amurense* Du	Heilongjiang Province, China	Aptian-Albian (Early Cretaceous); Coniacian-Maastrichtian (Late Cretaceous); Paleocene	Wood	Du, 1982 [54]; Wang et al., 1997 [55]; Terada et al., [56]
*Protopiceoxylon chaoyangensis* Duan	Liaoning Province, China	Aptian-Albian (Early Cretaceous)	Wood	Duan, 2000 [57]
*Protopiceoxylon yizhouensis* Duan et Cui	Liaoning Province, China	Aptian-Albian (Early Cretaceous)	Wood	Duan et al., 1995 [58]
*Keteleerioxylon arcticum* Shilkina	Franz Josef Land, Russia	Aptian-Albian (Early Cretaceous)	Wood	Shilkina, 1960 [7]
*Keteleerioxylon kamtschatkiense* Blokhina et Afonin	Kamchatka Peninsula, Russia	Aptian-Albian (Early Cretaceous); Turonian-Coniacian (Late Cretaceous)	Wood	Blokhina et al., 2006 [11]
*Keteleerioxylon changchunense* Shi, Sun, Meng et Yu	Jilin Province, China	Albian (Early Cretaceous)	Wood	This paper
*Protopiceoxylon johnseni* (Schroeter) Edwards	Svalbard, Norway	Early Cretaceous	Wood	Schröeter, 1880 [59]; Edwards, 1925 [60]
*Protopiceoxylon mohense* Ding	Heilongjiang Province, China	Early Cretaceous	Wood	Ding, 2000 [61]
*Keteleeria fortunei* (Andr. Murray) Carr.	Henan Province, China	Late Cretaceous	Wood	Yang et al., 1990 [39]
*Keteleeria cretacea* Miki et Maeda	Awaji, Japan	Late Cretaceous	Cone	Miki and Maeda, 1966 [62]
*Protopiceoxylon yukonense* Dolezych et Reinhardt	Yukon, Canada	Paleocene	Wood	Dolezych and Reinhardt, 2015 [63]
*Keteleeria* sp.	British Columbia, Canada	Early Eocene	Seed	Mathewes et al., 2016 [6]
*Keteleeria* sp.	Liaoning Province, China	Eocene	Cone	The Writing Group of Cenozoic Plants of China, 1978 [64]
*Keteleeria mabetiensis* Watari	Ishikawa Prefecture and Akita Prefecture, Japan	Oligocene and Miocene	Wood	Watari, 1941, 1956 [37,38]; Terada, 1998 [65]; Choi et al., 2010 [66]
*Keteleeria* sp.	Primorye, Russia	Oligocene to Early Miocene	Leaves, cones, and seeds	Rybalko et al., 1980 [67]
*Keteleerioxylon primoryense* Blokh.	Primorye, Russia	Oligocene to Miocene	Wood	Blokhina and Klimova, 2000 [35]
*Keteleeria rujadana* Lakhanpal	Oregon, USA	Oligocene	Cone	Lakhanpal, 1958 [68]
*Keteleeria ptesimosperma* Meyer et Manchester	Oregon, USA	Oligocene	Winged seeds	Meyer and Manchester, 1997 [69]
*Keteleeria prambachensis* (Hofmann) Klaus	Prambachkirchen, Austria	Oligocene	Cone	Hofmann, 1944 [70]; Klaus, 1977 [71]
*Keteleeria rhenana* Kräusel	Mainz, Germany	Early Miocene	Seed	Kräusel, 1938 [72]
*Keteleeria microreticulata* Ananova	Taganrog peninsula, Russia	Middle Miocene	Pollen	Ananova, 1974 [73]
*Keteleeria caucasica* Ramischvili	Zugdidi municipalitet, Georgia	Late Miocene	Pollen	Ramischvili, 1969 [74]
*Keteleeria davidiana* Miki	Honshu, Japan	Miocene	Cones and seeds	Miki 1941, 1957, 1958 [75,76,77]
*Keteleeria ezoana* Tanai	Niigata and Hokkaido Prefecture, Japan	Miocene	Seed scales, seeds, and leaves	Tanai, 1961 [78]; Tanai & Suzuki, 1963 [79]; Kamoi et al., 1978 [80]; Ozaki, 1979 [81]
*Keteleeria shanwangensis* Wang et al.	Shandong Province, China	Miocene	Cone, winged seeds,	Wang et al., 2006 [82]
*Keteleeria hoehnei* Kirchheimer	Saxony, Germany	Miocene	Cones, seeds, and needles	Kirchheimer, 1942 [83]; Kunzmann and Mai, 2005 [84]
*Keteleeria bergeri* Kirchheimer	Saxony, Germany	Miocene	Cones, seeds, and needles	Kirchheimer, 1942 [83]
*Keteleeria bergeri* Kirchheimer	Turowo, Poland; Kanton Schwyz, Switzerland	Miocene	Cone	Zalewska, 1961 [85]; Hantke, 1973 [86]
*Keteleeria heterophylloides* (Berry) Brown	Idaho, USA	Miocene	Vegetative shoots	Brown, 1935 [87]
*Keteleeria zhilinii* Blokh. et Bondarenko	Primorye, Russia	Pliocene	Wood	Blokhina and Bondarenko, 2005 [10]
*Keteleeria* sp.	Maoming, China	Late Pleistocene	Wood	Huang et al., 2019 [12]
*Keteleeria* sp.	Wuhan, China	Early Holocene	Wood	Yang et al., 2003 [40]

## Data Availability

The data presented in this study are available on request from the corresponding author.

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
