# Peer review of "Early Cretaceous Keteleerioxylon Wood in the Songliao Basin, Northeast China, and Its Geographic and Environmental Implications"

_biology, 2022, doi:10.3390/biology11111624_

Round 1
Reviewer 1 Report
The manuscript by Shi and co-authors present a novel wood fossil of Keteleerioxylon changchunense Shi, Sun, Meng et Yu sp. nov. collected from the middle member of Yingcheng Formation, Yingcheng Coal Mine, Changchun City, Jilin Province, Northeast China. The new wood fossil of Keteleeria, Keteleerioxylon changchunense, has been described in the Early Cretaceous strata about 110 million years ago in Changchun, Jilin Province, Northeast China. The quantitative growth-ring analyses of K. changchunense indicate that it was evergreen and its leaf longevity was 1-3 years. Its high ring markedness index (RMI) indicates that the climate seasonality was pronounced during the early Albian in Songliao Basin, Northeast China.
The manuscript fit well with the standards of Biology. The text is logically structured and easy to read. The methodological approach of quantitative analysis, accurately described by the authors, is sound. The results and discussion are accurately presented also with the help of necessary high-quality figures and tables.
I think that the study deserves publication in Biology.
Author Response
Thanks for your comment on the manuscript.
Reviewer 2 Report
The authors describe a silicified wood from the Lower Cretaceous of the Jilin Province (China). They assign it to the genus Keteleerioxylon Shilkina. From the analysis of the growth rings they derive palaeoecological hypotheses. From the supposed affinity with Keteleeria they derive a palaeobiogeographical discussion.
The manuscript is formally well prepared. Except that in several places the English is confusing, even wrong, and even locally suggests that the authors thought they were saying something other than what they were saying. The use of tenses is particularly confusing.
The illustration is of good quality. However, Figure 7 is largely redundant with Figure 6.
The structure of the ms needs to be partially revised. What is a distribution map of current Keteleeria species doing in the introduction - when it is not used in the problematic?
On the borderline between form and content, on which sample is the description based? How do the two samples mentioned differ? There is also a probable error in this description, as Figure 3G does not illustrate triseriate intertracheary pits.
On the substantive side there are many problems. The authors use a concept, the "Keteleeria-type", which they do not define anywhere. Their attribution to the genus Keteleerioxylon is based on the single argument "The fossil woods described here ... closely resemble Keteleeria, Keteleerioxylon and Protopiceoxylon", which is short. The authors ignore the fact that Shilkina's original diagnosis specifies the presence of a part of alternate intertracheary pits. Several times they point out the similarity with Protopiceoxylon but nowhere do they say why they rather retained Keteleerioxylon. They ignore the fact that Pinoxylon Knowlton is a taxonomic synonym of Protopiceoxylon Gothan (Medlyn & Tidwell, 1979). They do not seem to know about Keteleerioxylon liaoxiense (Jiang et al., 2019). Nor do they seem to know the Pinoxylon described from north-east Asia, such as P. yabei and probably several other Protopiceoxylon. All this suggests that, like too many authors, they first decided that their wood was Keteleeria wood, and then adapted their nomenclatural and taxonomical approaches, an extremely perverse practice that has generated an abominable mess in palaeoxylology. In fact there are dozens of such woods reported from the Cretaceous and Tertiary of the northern hemisphere that are close to the described ones and are not considered for discussion because they are described as Piceoxylon, Tsugoxylon, Abietoxylon, etc.
Finally, it seems incredible that in 2022 the authors, while discussing their results, did not cross-reference their own with those of molecular phylogeny. They would have seen (e.g. with Ran et al. Molecular Phylogenetics and Evolution 129 (2018) 106-116) that Keteleeria and Abies separated late in the Tertiary, and that a (supposed) anatomical similarity with Keteleeria for a Cretaceous wood cannot be characteristic of the presence of exclusive ancestors of this genus, especially since the fossil woods affine to the Pinaceae show a puzzling variability within even a single specimen.

Author Response
Please check the reply in the attachment.

Reviewer 3 Report
The manuscript of entitled “Early Cretaceous Keteleerioxylon Wood in Songliao Basin, Northeast China, and Its Geographic and Environmental Implications" by Shi et al. is an impressive paper dealing with the discovery of a new conifer wood fossil and its palaeogeographical and palaeoenvironmental indications. The anatomical details of the wood were well described and illustrated. The taxonomic assignment of the wood is reliable. Additionally, by reviewing Keteleeria and closely related fossil taxa, the authors also proposed the probable origin and migration route of Keteleeria-like group. Furthermore, the potential link between the migration of Keteleeria-like plants and paleoclimate variations was also discussed by the author. All these results are believed to be receivable.
Generally, I think that this paper is of great importance and worthy to be published in the Journal BIOLOGY.
However, I also give several comments as follows that need to be explained by the authors:
1.The presence or not of Abietineentüpfelung on the transverse (tangential) wall of ray cells is commonly believed to be valuable in the identification of Pinaceae wood. A detailed description of this character is suggested to be given by the authors.
2. The quantification of ring markedness parameters of the extant Keteleeria species is suggested to be included in Table 3, if possible.
3. Just as the authors mentioned above, the earliest Albian climate is regarded as a greenhouse, then why the seasonality is so distinct? An analysis of the palaeoaltitudinal implication of the present fossil wood is suggested.
4. There are also some minor amendments which are marked in the PDF of manuscript that need to be corrected by the authors.

Author Response
Thank you for your constructive comments. I add photos of the characteristic Abietineentüpfelung in the manuscript. For the ring markedness parameters of the extant Keteleeria species, since the extant Keteleeria live in monsoon climates, it is meaningless to calculate the ring markedness parameters. As the paleoclimate indication of the wood, we think that the temperature is only one aspect of seasonality, and precipitation, rather than elevation, may be an important factor in affecting annual rings, but it is difficult to discuss further based on current materials. Thanks for the reviewer improving the English. We modify in the manuscript.